# Predictors of time to death among under-five children in pastoral regions of Ethiopia: A retrospective follow-up study

**Bsrat Tesfay Hagos**[1]☺*, **Gebru Gebremeskel Gebrerufael**[2]☺

**1** Department of Statistics, College of Natural and Computational Science, Mekelle University, Mekelle, Ethiopia, **2** Department of Statistics, College of Natural and Computational Science, Adigrat University, Adigrat, Ethiopia

☺ These authors contributed equally to this work.
* bsrattesfaygu@gmail.com

**Data Availability Statement:** Datasets cannot be shared publicly because they contain sensitive participant information. Datasets are available from the EDHS after approval from the IRB unit. Access

## Abstract

### Background

In Ethiopia, the mortality rate for children under five is a public health concern. Regretfully, the problem is notably underestimated and underreported, making it impossible to fully recognize how serious the situation is in the nation's developing regions. Unfortunately, no single study has been conducted to reveal the rates and predictor factors of under-five child death in Ethiopia's pastoral regions. Therefore, the purpose of this study was to determine the critical variables that led to a shorter survival time to death for children in Ethiopia's pastoral regions under the age of five.

### Methods

Between January 18 and June 27, 2016, a retrospective follow-up study was done among under-five children in pastoral areas of Ethiopia. The statistically significant difference between categorical predictors was shown using the log-rank test, and the Kaplan-Meier survival curve was used to determine the survival time. In order to identify the time-to-death predictor factors in children under five, Cox proportional hazards (PH) model analyses of bivariable and multivariable variables were fitted.

### Results

A total 7,677 children were included in the study. The overall incidence rate of under-five mortality was 8.4% (95% CI 7.77%, 9.0%). In the multivariable Cox PH model analysis, children vaccinated (AHR: 0.72, 95% CI: 0.59, 0.88), mothers aged 35–40 (AHR: 1.27; 95% CI: 1.06, 1.52), and above 41 (AHR: 2.18, 95% CI: 1.59, 2.98), not initiating exclusively breastfeeding (AHR: 1.26, 95% CI: 1.02, 1.55), the agriculture sector of the mother's occupation (AHR: 2.57, 95% CI: 1.74, 3.31), the male sex of the household head (AHR: 0.67, 95% CI: 0.56, 0.81), non-anemic child (AHR: 0.67, 95% CI: 0.55, 0.83), and rural residence (AHR: 3.27, 95% CI: 1.45, 7.38) were identified as main predictors of time to death among under-five children.

to the study data set requires legal registration at https://dhsprogram.com/data/available-datasets.cfm and the creation of a persuasive letter outlining the project descriptive goal for the DHS program. More access information can also be found on the DHS Program website (https://dhsprogram.com/data/Access-Instructions.cfm). The authors confirm that interested researchers would be able to access these data in the same manner as the authors. The authors also confirm that they had no special access privileges that others would not have.

**Funding:** The authors received no specific funding for this work.

**Competing interests:** The authors have declared that no competing interests exist.

**Abbreviations:** AHR, Adjusted Hazard Ratio; ANC, Ante-Natal Care; CHR, Crude Hazard Ratio; CI, : Confidence Interval; CSA, Central Statistical Agency; EDHS, Ethiopia Demographic Health Survey; KM, Kaplan-Meier; LRT, Likelihood Ratio Test; MDG4, Fourth Millennium Development Goal; MOH, : Minister of Health; PH, Proportional-Hazards; SSA, Sub-Saharan African; U5CMR, Under-5 Children Mortality Rate.

## Conclusions

In this study, the authors found a higher rate of under-five deaths than the national figure. A child vaccinated, exclusively breastfeeding, mother's occupation, sex of household head, anemic child, mother's age, and residence were found to be the most influential predictors for time-to-death. Therefore, to lower the high incidence of under-five mortality, the government should focus on the pastoral regional states of Ethiopia.

## Introduction

The likelihood that a child born in a given year will pass away before turning five is known as the under-5 child mortality rate (U5CMR). Children's intellectual and physical development is crucial throughout the first five years of life [1]. Malnutrition continues to be a major global public health issue and a significant cause of child morbidity and mortality [2].

Although the U5CMR has decreased globally, from 5.9 million child deaths in 2015 to 5.3 million in 2018, there is still a high mortality rate in African nations, with Ethiopia having the highest rate at nearly one-twelfth (81 deaths per 1,000 live births), which is roughly seven times higher than in European nations [1].

More than 25% of all under-five-year-old child deaths worldwide take place in Africa [3]. Somalia (133/1000 live births), Chad (127/1000 live births), the Central African Republic (124/1000 live births), Sierra Leone (114/1000 live births), Mali (111/1000 live births), and Nigeria (104/1000 live births) were the top six African nations with the highest U5CMR in 2016, accounting for more than 50% of under-five mortality in Sub-Saharan African (SSA) countries and roughly 1/5 of under-five mortality globally [3].

With one child out of every eleven live births dying before the age of five, the SSA continues to have the highest U5CMR in the world [4, 5]. So, the death of children under the age of five continues to be a major public health issue in the SSA [5]. Similar to this, U5CMR in Ethiopia is one of the most significant and challenging public health issues that need to be highlighted [6].

The 2016 Sustainable Development Goals (SDG) have included strategies and measures that have the potential to considerably lower U5CMR below 25 per 1000 live births overall. A few undeveloped countries have requested a decline in under-five mortality; still, sub-Saharan African low- and middle-income countries continue to have extremely high U5CMR [7].

With Ethiopia's current child mortality rates, it will be difficult to achieve the 2030 goal of reducing the U5CMR to 55 deaths per 1,000 live births, and research has shown that there is a significant regional difference in under-five mortality [6]. The highest rate was recorded in Somalia, Benishangul-Gumuz, and Afar national regional states, where it was estimated to be about 94, 98, and 125 deaths per 1,000 live births, respectively [8]. This means that the U5CMR in Ethiopia is still higher (i.e., reported to be 67 deaths per 1,000 live births). Additionally, research from many sources showed that among the predictor factors linked to U5CMR were anemic children, poor mother education level, failure to initiate breastfeeding, antenatal care (ANC) fellows, advanced mother age, and unvaccinated children [1, 3, 9–11]. To alleviate these problems, Ethiopia is one of the SSA nations working to reduce U5CMR. The government has implemented various interventions to improve coverage, quality, and use of skilled care, as well as important newborn care and management of preterm and low birth weights. However, it continued to be quite high in Ethiopia's pastoral regions, including Somalia, Benishangul-Gumuz, and Afar [7, 8]. Additionally, local studies carried out in developed

regions of the country indicated significant subnational variations in mortality rates and causes, which might be concealed by estimation methods at the national level. Therefore, this study was aimed to assess the potential predictor factors that led to a shorter survival time-to-death for children under the age of five in pastoral regions of Ethiopia. The data from this study will serve as a foundation for further research, and the results will aid in the development of programs to lengthen the survival period of children under five in the study area and other regions of Ethiopia. Furthermore, this research could provide insights into the best ways to reduce mortality among children under five for medical professionals, policymakers, and the general public.

## Materials and methods

### Study design, source of data, and period

This retrospective follow-up study design was conducted using secondary data analysis based on the 2016 Ethiopia Demographic and Health Survey (EDHS) data. A nationally representative sample of roughly 16,650 homes was chosen for the 2016 EDHS, which was carried out with the cooperation of the Ministry of Health (MOH) and carried out by the Central Statistical Agency (CSA) and partner organizations between January 18, 2016, and June 27, 2016. Individual interviews were available with all of the men and women in these families who were of reproductive age (15–49 and 15–59, respectively).

### Sampling technique, inclusion, and exclusion criteria

A stratified sample design with two stages was used by the 2016 EDHS. Nine regional states and two administrative cities make up the nation. 443 rural and 202 urban enumeration areas were chosen at random in the first round, with the probability proportionate to size. In the second stage, a cluster of equal probability and systematic selection from the freshly compiled household list involved 28 homes. The regional states of Afar, Benishangul-Gumuz, and Somalia were chosen because, according to the 2016 EDHS data, they have the highest U5CMR. In these areas, a total sample of 7,677 women with live births in the five years before the survey (642 occurrences and 7,035 censored children) was included for this study's analysis. The detailed diagrammatic sample selection procedure was presented (see **Fig 1**).

### Study variables

**Response variable.**   The child's time to death, which was measured in months from birth to the end of the follow-up period, was the response variable. Events were defined as child deaths between birth and 59 months (1 = died). Children who were still alive at the end of the follow-up period and had not turned five were classified as censored (0 = censored).

**Predictor variables.**   The primary factors connected with time-to-death among children under the age of five are type of birth child's (single, multiple), father's education level (no education, primary, secondary and above), mother's occupational (not working, non-agriculture sector, agriculture sector), region (Affar, Somalia, Benshangul-gumuaz), residence (urban, rural), toilet facilities (no facility, with facility), mother's education level (no education, primary and above), sex of household head (female, male), sex of child (male, female), ante-natal care (ANC) fellow-up (no, yes), modern contraceptive method used (no, yes), anemic child (yes, no), exclusively breastfeeding (yes, no), mother's age (in year) (34 and below, 35–40, 41 and above), and child vaccinated (no, yes); they were chosen for this investigation from the available similar studies on the topic.

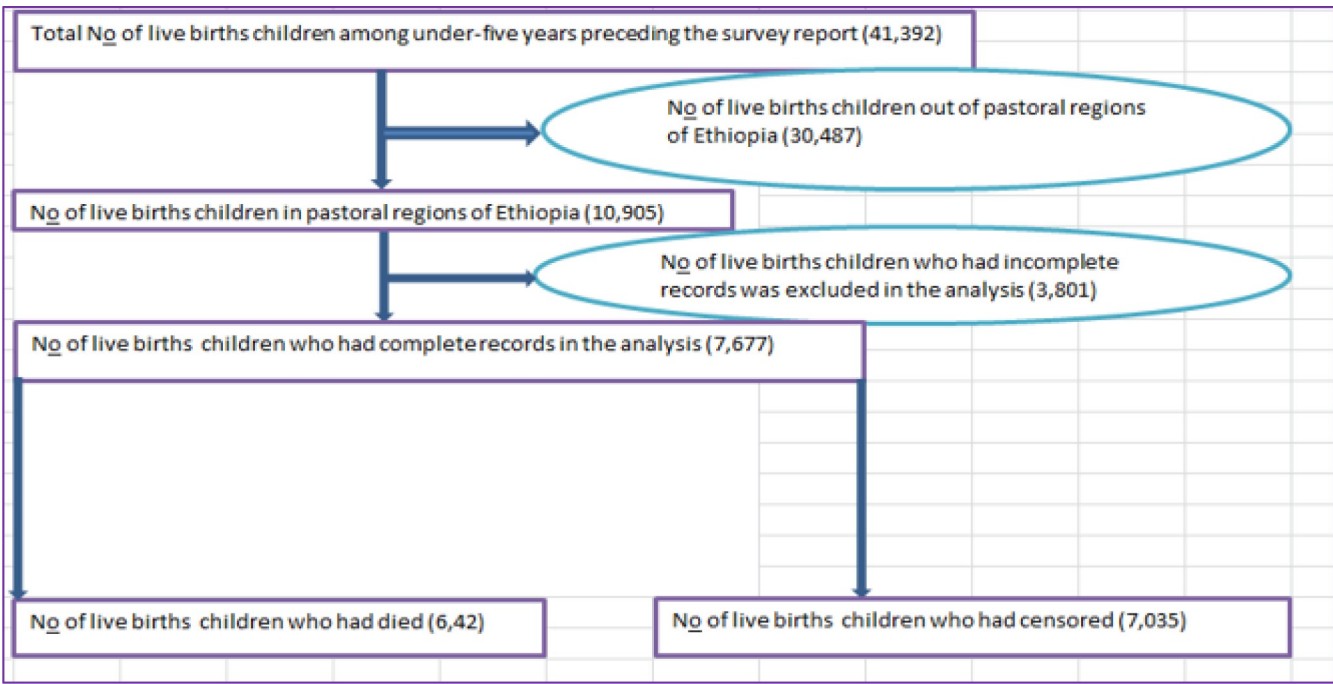

**Fig 1. Diagrammatic presentation of sample selection among under-five children included in this study, 2016.**

**Statistical methods of data analysis.** In the bivariable STATA software version 14 package (i.e., the data was extracted, recorded, and analyzed). To explore all of the study's variables, descriptive statistics, including frequency distributions and percentages, were used. For categorical predictor variables, the Kaplan-Meier (K-M) estimators were used to display the survival experience of children through time. In order to determine whether there is a statistically significant difference between the predictor variables contributing to the ability of under-five children to survive, the survival curves were compared and tested using the log-rank test (see S1 Table). The Cox proportional hazards (PH) model is one of the regression models used in survival analysis most frequently. Modeling Cox proportional hazards (PH),

$h(t) = h_0(t)*exp^{(b_1x_1+b_2x_2+b_3x_3+\ldots\ldots\ldots b_px_p)}$ Were used to identify and check the impact of each independent variable on the time-to-death. The hazard function $h(t)$ is determined by a set of p covariates $(x_1, x_2, x_3, \ldots\ldots, x_p)$, whose impact is measured by the size of the respective coefficients $(b_1, b_2, b_3, \ldots\ldots, b_p)$. The term $h_0(t)$ is called the baseline hazard for a time-to-death. All predictor variables and the overall proportionality test (Shoenfield residual test) provided evidence in support of the proportionality assumptions of the Cox PH regression model analysis. Each predictor variable's bivariable Cox PH regression model was fitted once the proportional hazard assumption had been verified.

Furthermore, those variables with a p-value $\leq 0.25$ in the bivariable model analysis were included in the multivariable Cox PH regression model. Crude and adjusted hazard ratios with a 95% confidence interval (CI) were utilized to examine statistically significant factors and gauge the strength of associations. In the multivariable Cox PH regression model analysis, variables with a P-value $< 0.05$ were discovered to be significant indicators of time to death in children under the age of five. The Likelihood Ratio Test (LRT) was used to evaluate the final model's goodness of fit.

**Measurements.** In this study, "**survival time**" was defined as the period from the commencement of observation to the occurrence of the observation's result (event or suppression).

**Event.** In this study, the event referred to the study subject (1), who had died during the observation period and had experienced the event outcome.

**Censored.** The study of children under the age of five who were still alive at the end of the follow-up is referred to as censored.

### Ethical consideration

In order to use the 2016 EDHS dataset from the DHS program for the current study, a formal consent letter was discovered. The dataset was also given IRB approval for public use without individual or household identity. As a result, the participants' identities were kept secret. Furthermore, this dataset was exclusively used for the current study in accordance with the DHS program discipline, rules, and regulations. However, access to the study's data set requires legal registration at https://dhsprogram.com/data/available-datasets.cfm and the creation of a persuasive letter outlining the project's descriptive goal for the DHS program.

## Results

### Descriptive statistics

In this study, a total sample of 7,677 was selected for analysis, with 642 under-five deaths. The majority (96.6%) and (84.7%) of the study participants lived in rural areas and had no education level, respectively. Approximately 98.6% of study participants were single-birth types, and 49.2% were from the Somali regional state. About 79.1% of respondents had toilet facilities, and 64.2% of participants were female household heads. Concerning the use of the modern contraceptive method, 94.1% of respondents did not use the modern contraceptive method.

Most (75.6%) fathers have no education level, and 64.3% of mothers did not have ANC follow-up during their last pregnancy. Moreover, 51.9% of children were male, and 79.1% of children were not anemic (**Table 1**).

### Comparison of the survival ability of under-five children

Along with the descriptive data shown in **Table 2** above, Kaplan-Meier (K-M) survival curve estimators are shown in **Figs 2–7** for the most important predictor variables. These samples of K-M curves show that under-five-year-old children whose advanced mother's age, rural residence, being male sex household head, being anemic, not vaccinated, and not initiating exclusive breastfeeding had shorter survival times compared to those reference categories. Additionally, from the log-rank test, there was a significant difference between the mother's age (P-value = 0.001), residence groups (P-value = 0.001), sex of household head (P-value = 0.001), anemic child (P-value = 0.001), vaccinated child (P-value = 0.001), and exclusive breastfeeding (P-value = 0.001) on the time-to-death among under-five children (see **S1 Table**).

### Time-to-death predictors among under-five children

According to this study, there were 8.4 deaths per 100 live births over the study period as a general result of U5CMR. The mean follow-up period was 9 months. None of the predictors violated the Cox PH regression model's essential assumption (see **Table 2**).

Significant predictors of U5CMR in the bivariable Cox PH regression model study (p-value ≤ 0.25): type of birth, father's education level, region, mother's occupation, residence, toilet facilities, mother's education level, sex of household head, sex of the child, ANC fellow-up, modern contraceptives used, anemic child, exclusively breastfeeding, mother's age, and child vaccinated. Only eight variables were shown to be highly significant predictors of time to death in under-five-year-old children in the multivariable Cox PH regression model study. As

**Table 1. Socio-demographic characteristics and environmental predictors in pastoral regions, Ethiopia, 2016 (N = 7,677).**

| Variables | Categories | Frequency (N) | Percentage (%) |
|---|---|---|---|
| Type of birth | Single | 7572 | 98.6 |
| | Multiple | 105 | 1.4 |
| Father's education level | No education | 5803 | 75.6 |
| | Primary | 1086 | 14.1 |
| | Secondary and above | 788 | 10.3 |
| Mother's occupation | Not working | 5722 | 74.5 |
| | Non-agriculture sector | 1680 | 21.9 |
| | Agriculture sector | 275 | 3.6 |
| Region | Affar | 2287 | 29.8 |
| | Somalia | 3775 | 49.2 |
| | Benshangul-gumuaz | 1615 | 21 |
| Residence | Urban | 261 | 3.4 |
| | Rural | 7416 | 96.6 |
| Toilet facilities | No facility | 1606 | 20.9 |
| | With facility | 6071 | 79.1 |
| Mother's education level | No education | 6502 | 84.7 |
| | Primary and above | 1175 | 15.3 |
| Sex of household head | Female | 4931 | 64.2 |
| | Male | 2746 | 35.8 |
| Sex of child | Male | 3986 | 51.9 |
| | Female | 3691 | 48.1 |
| ANC fellow-up | No | 4937 | 64.3 |
| | Yes | 2740 | 35.7 |
| Modern contraceptive used | No | 7224 | 94.1 |
| | Yes | 453 | 5.9 |
| Anemic Child | Yes | 1605 | 20.9 |
| | No | 6072 | 79.1 |
| Exclusively breastfeeding | Yes | 4738 | 61.7 |
| | No | 2939 | 38.3 |
| Mother's age | 34 and below | 3363 | 43.8 |
| | 35–40 | 3780 | 49.2 |
| | 41 and above | 534 | 7 |
| Child vaccinated | No | 3058 | 39.8 |
| | Yes | 4619 | 60.2 |

a result, the adjusted hazard ratios (AHR) for children born to mothers who were older than average were 1.27 and 2.18 times higher than those for children born to younger mothers (AHR = 1.27; 95% CI: 1.06, 1.52) and (AHR = 2.18; 95% CI: 1.59, 2.98), respectively. The risk of death for children born to mothers who worked in the agricultural sector was 2.57 times higher than the risk for children born to mothers who did not work (AHR: 2.57; 95% CI: 1.74, 3.31). Children from a male household head had under-five mortality rates that were nearly 33% lower than those from a female household head (AHR: 0.67; 95% CI: 0.56, 0.81) for the variable sex of household head. Children who had not been anemic were 33% less likely to die (AHR: 0.67; 95% CI: 0.55, 0.83) than children who had been anemic. Additionally, compared to children who exclusively breastfed from the beginning, children who did not were 1.26 times (AHR: 1.26; 95% CI: 1.02, 1.55) more likely to pass away. Furthermore, a strong correlation between child immunization and time to death in children under the age of five was

**Table 2. Test of proportional-hazards assumption (STATA software version 14).**

| Variables | Rho | Chi$^2$ | P-value |
|---|---|---|---|
| Type of birth (reff. = single) | | | |
| Multiple | -0.024 | 0.3 | 0.5808 |
| Father's education level (reff. = no education) | | | |
| Primary | 0.0537 | 1.46 | 0.2275 |
| Secondary and above | -0.024 | 0.29 | 0.5923 |
| Mother's occupation (reff. = not working) | | | |
| Non-agriculture sector | 0.07202 | 2.76 | 0.0964 |
| Agriculture sector | -0.0201 | 0.23 | 0.6347 |
| Region (reff. = Affar) | | | |
| Somalia | -0.01 | 0.06 | 0.8087 |
| Benshangul-gumuaz | -0.028 | 0.48 | 0.4884 |
| Residence (reff. = urban) | | | |
| Rural | -0.068 | 2.68 | 0.1018 |
| Toilet facilities (reff. = no facility) | | | |
| With facility | 0.029 | 0.55 | 0.4597 |
| Mother's education level (reff. = no education) | | | |
| Primary and above | -0.047 | 1.23 | 0.2675 |
| Sex of household head (reff. = female) | | | |
| Male | -0.006 | 0.02 | 0.8878 |
| Sex of child (reff. = male) | | | |
| Female | -0.04 | 0.86 | 0.3527 |
| ANC fellow up (reff. = no) | | | |
| Yes | 0.0291 | 0.48 | 0.4876 |
| Modern contraceptive used (reff. = no) | | | |
| Yes | -0.036 | 0.75 | 0.3874 |
| Anemic Child (reff. = yes) | | | |
| No | -0.023 | 0.28 | 0.5944 |
| Exclusively breastfeeding (reff. = yes) | | | |
| No | -0.033 | 0.62 | 0.4321 |
| Mother's age (reff. = 34 and below) | | | |
| 35–40 | 0.0744 | 3.02 | 0.0822 |
| 41 and above | 0.04422 | 1.07 | 0.3019 |
| Child vaccinated (reff. = no) | | | |
| Yes | 0.03674 | 0.76 | 0.3832 |
| Global test | ————— | 18.56 | 0.4854 |

**N.B:** Chisq refers to chi-square statistic value; Rho refers to the correlation among covariates and survival times.

discovered. Children who received vaccinations had under-five time-to-death risks that were 38% (AHR: 0.72; 95% CI: 0.59, 0.88) lower than children who did not receive vaccinations. Finally, the residency had a negative effect on the time to death for children under the age of five. Children whose mothers resided in rural areas had 3.27 times (AHR: 3.27; 95% CI: 1.45, 7.38) greater mortality rates than children whose mothers resided in urban areas (**Table 3**).

## Cox PH regression model and model adequacy checking results

A study of the Cox PH regression model was used to determine the impact of each covariate on the time until a child under the age of five died. The Likelihood Ratio Test (LRT) of the

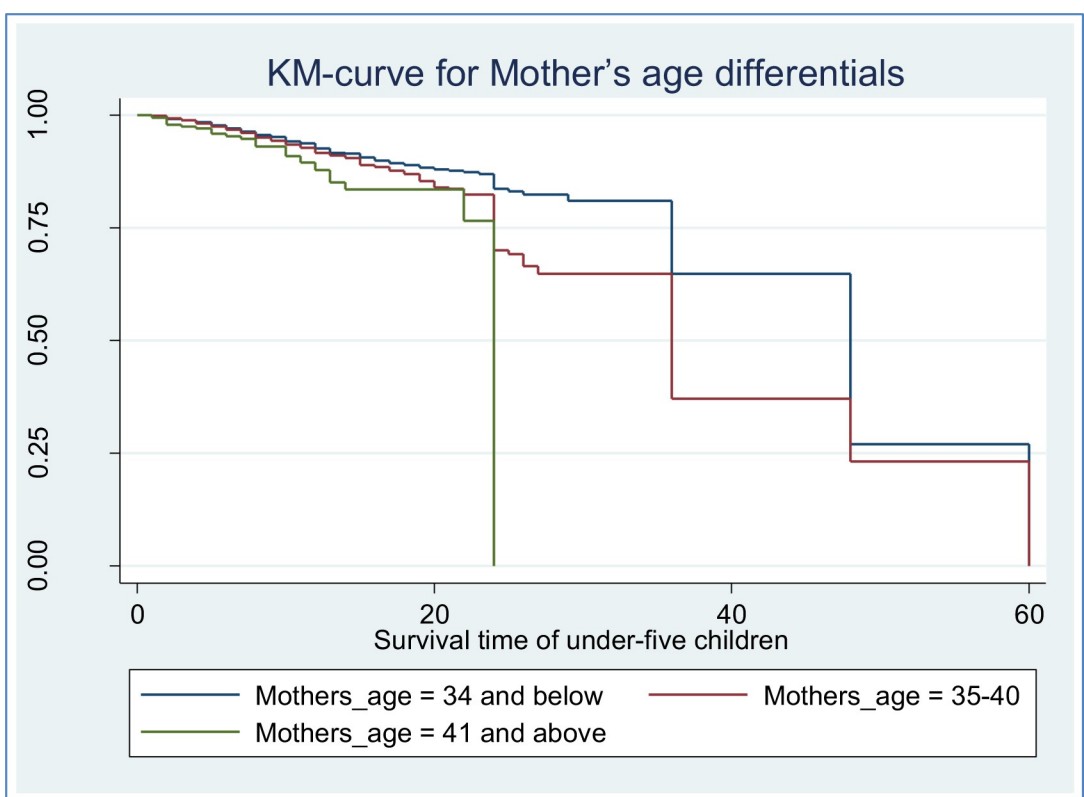

**Fig 2. Survival curve by mothers age of children.**

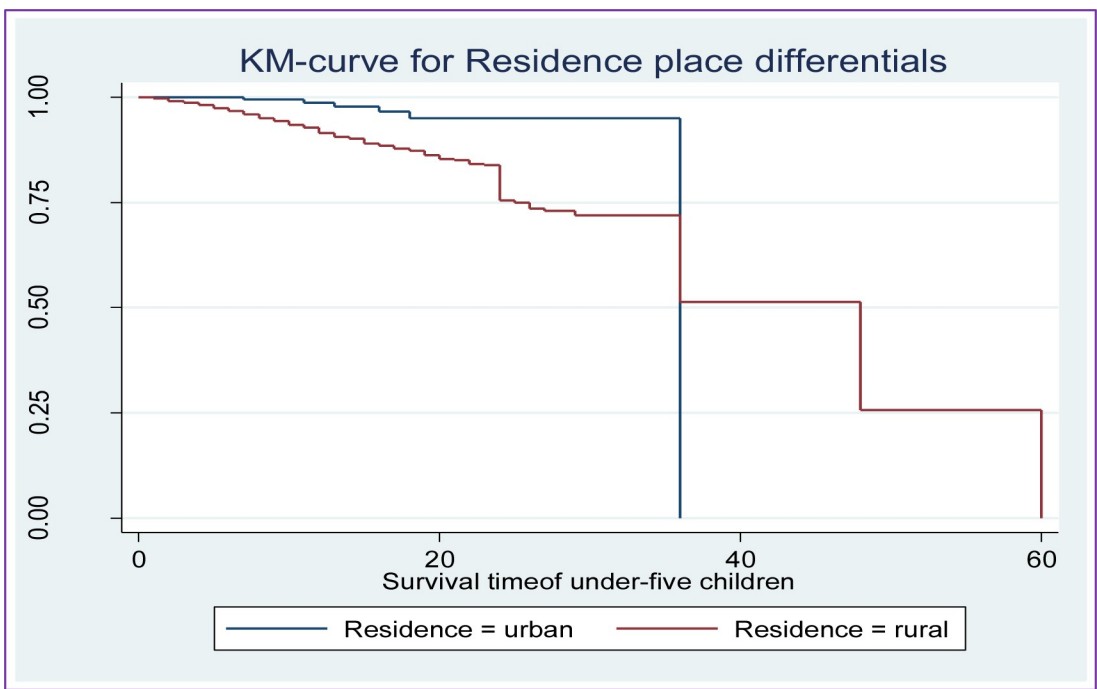

**Fig 3. Survival curve by residence place of children.**

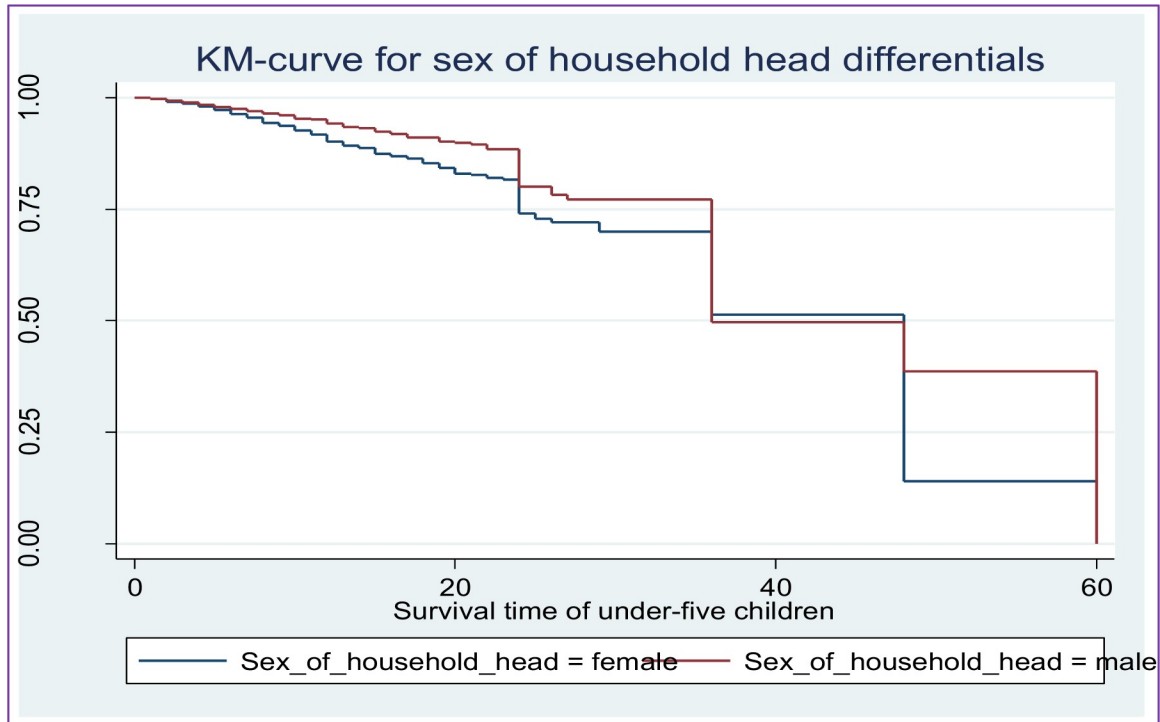

**Fig 4. Survival curve by sex of household head.**

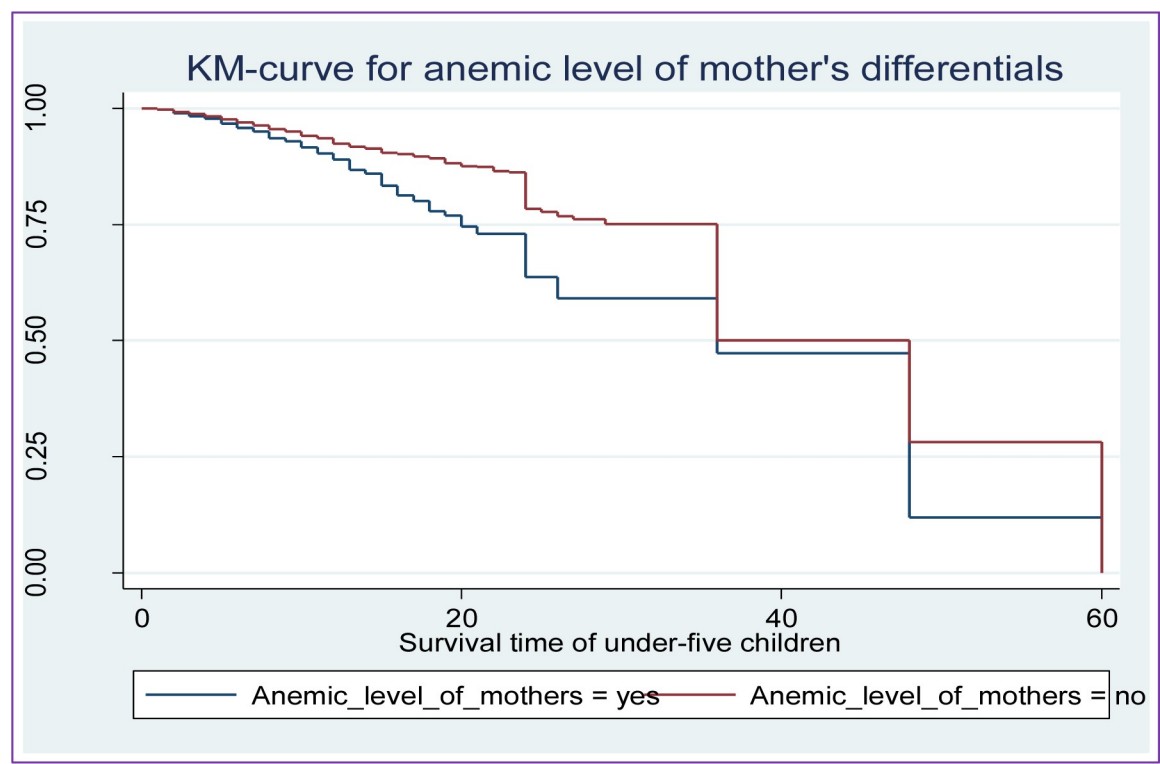

**Fig 5. Survival curve by anemic level of mother's.**

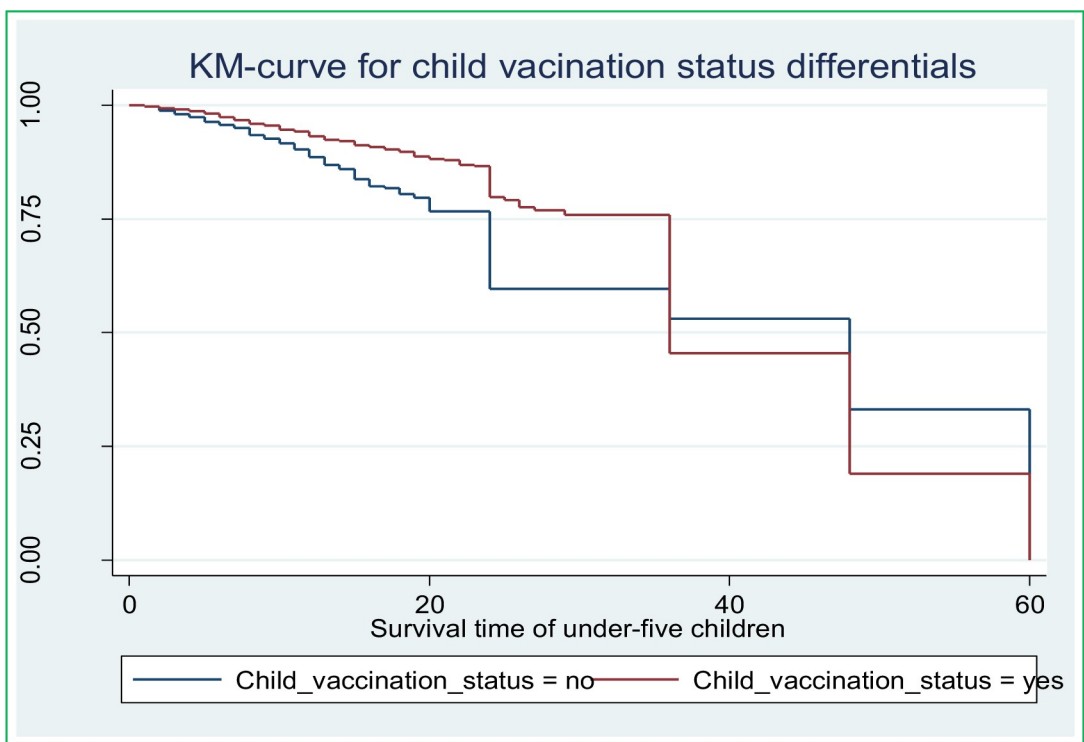

**Fig 6. Survival curve child vaccination.**

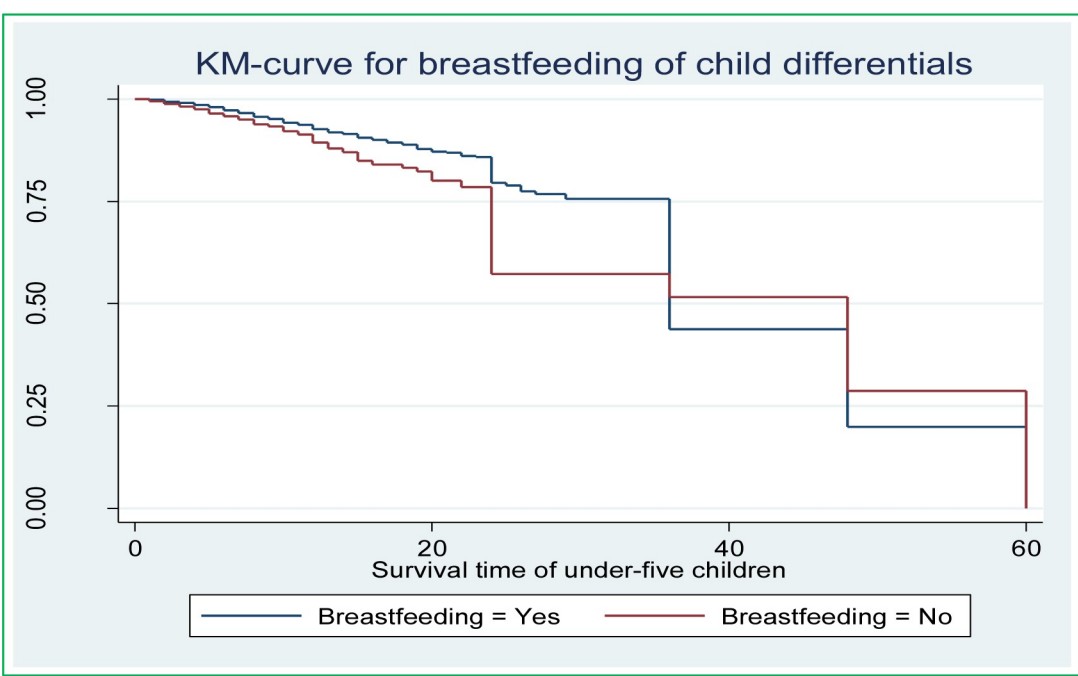

**Fig 7. Survival curve by breastfeeding of child.**

**Table 3. Bivariable and multivariable Cox PH regression model analysis of time-to-death predictors among under-five children in the pastoral regions of Ethiopia, 2016 (N = 7,677).**

| Variables | Survival status | | CHR (95% CI) | AHR (95% CI) | P-value |
|---|---|---|---|---|---|
| | Censored | Event | | | |
| Mother's occupation | | | | | |
| Not working (reff.) | 5263 | 459 | 1 | 1 | |
| Non-agriculture sector | 1547 | 133 | 0.64 (0.52, 0.8)* | 0.78 (0.63, 0.98)* | 0.036* |
| Agriculture sector | 225 | 50 | 2.1 (1.54, 2.86)* | 2.57 (1.74, 3.31)* | 0.001* |
| Residence | | | | | |
| Urban (reff.) | 250 | 11 | 1 | 1 | |
| Rural | 6785 | 631 | 3.88 (1.74, 8.69)* | 3.27 (1.45, 7.38)* | 0.004* |
| Sex of house hold head | | | | | |
| Female (reff.) | 4494 | 437 | 1 | 1 | |
| Male | 2541 | 205 | 0.65 (0.54, 0.78)* | 0.67 (0.56, 0.81)* | 0.007* |
| ANC fellow up | | | | | |
| No (reff.) | 4490 | 447 | 1 | 1 | |
| Yes | 2545 | 195 | 0.84 (0.70, 1.01) | 0.89 (0.73, 1.08)* | 0.231 |
| Anemic Child | | | | | |
| Yes (reff.) | 1452 | 153 | 1 | 1 | |
| No | 5583 | 489 | 0.61 (0.50, 0.74)* | 0.67 (0.55, 0.83)* | 0.001* |
| Exclusively breastfeeding | | | | | |
| Yes (reff.) | 4310 | 428 | 1 | 1 | |
| No | 2725 | 214 | 1.52 (1.26, 1.83)* | 1.26 (1.02, 1.55)* | 0.030* |
| Mother's age | | | | | |
| 34 and below | 3100 | 263 | 1 | 1 | |
| 35–40 | 3459 | 321 | 1.29 (1.08, 1.55)* | 1.27 (1.06, 1.52)* | 0.010* |
| 41 and above | 476 | 58 | 2.0 (1.147, 2.74)* | 2.18 (1.59, 2.98)* | 0.001* |
| Child vaccinated (reff. = no) | | | | | |
| No (reff.) | 2809 | 249 | 1 | 1 | |
| Yes | 4226 | 393 | 0.58 (0.48, 0.70)* | 0.72 (0.59, 0.88)* | 0.002* |

NB

*Significant at p-value < 0.05.

**Abbreviations:** reff.: reference

LRT = LR chi$^2$(10) = 142.32 (p < 0.001)

Cox PH regression model analysis with the Breslow method was used to test model adequacy. The test statistic was 142.32 with a *p*-value of 0.000, which showed that the model is a good fit for the data set at a 5% level of significance (**Table 3**).

## Discussion

Although there have been many advancements and measures to increase newborn survival, U5CMR is still a significant public health issue in the SSA, with Ethiopia being severely afflicted. Therefore, to ascertain U5CMR in Ethiopia's pastoral regions, the authors carried out this retrospective follow-up investigation. Accordingly, 84 deaths per 1,000 live births were reported to represent the overall incidence of U5CMR in Ethiopia's pastoral regions. This result is in line with a prior investigation in SSA (81 deaths per 1,000 live births) [3], Ethiopia (61 deaths per 1,000 live births) [1], and Nigeria (104 deaths per 1,000 live births). The authors results, however, are lower than those of other investigations carried out in Ethiopia (273

deaths per 1,000 live births), Somalia (133 deaths per 1,000 live births), Chad (127 deaths per 1,000 live births), the Central African Republic (124 deaths per 1,000 live births), Sierra Leone (114 deaths per 1,000 live births), Mali (111 deaths per 1,000 live births) [3], and Ghana (250 deaths per 1,000 live births) [10].

On the other hand, these results are significantly greater than the U5CMR incidence reported in northern Ethiopia (35.62 deaths per 1,000 live births) [5], the Somali region of Ethiopia (57 deaths per 1,000 live births) [12], the rural area of Eastern Ethiopia (28.37 deaths per 1,000 live births) [13], the Tigray region of northern Ethiopia (62.5 deaths per 1,000 live births) [14], and Ethiopia (36.7 deaths per 1,000 live births) [15]. The foregoing variables in study designs, sample sizes, study periods, study sites, and sociodemographic characteristics of study participants may help to explain the variances between studies. According to this study, children under the age of five who were not anemic had a 33% lower risk of dying than those who were. A study conducted in Ethiopia, Jimma University Specialized Hospital, Gedeo Zone, and Sekota Waghemra Zone of northern Ethiopia also demonstrated that severe anemia had a decreased case fatality rate [9, 16–18]. Similarly, research from South Africa and Niger [19, 20] confirmed that exposed children have a greater risk of dying before the age of five. This is due to the fact that anemia will result in less compliance overall as well as an increase in infection incidence and heart failure risk. The other rationale for this study's findings is that anemia in young children might result in growth retardation and preterm birth, which are the biggest risk factors for mortality among under-five-year-olds.

The hazards of time-to-death for under-five children were lower among male household heads than female household heads for the predictor variable, sex of household heads. Children under the age of five who came from moms with fewer overall children were at the lowest risk of dying [21–23].

Additionally, among children under the age of five, exclusively breastfeeding was a significant predictor of time to death. This research showed that, compared to their peers who exclusively breastfed, children who did not start breastfeeding were at a higher risk of dying. This discovery is in line with a study carried out in Ethiopia [1]. According to research done in other SSA nations, breastfeeding can reduce the risk of time-to-death in children under the age of five by 16% if it begins within the first day of delivery and by as much as 22% if it begins within an hour [24].

The time to death among children under the age of five was strongly correlated with the mother's occupation. As a result, babies born to moms who work in agriculture have a higher risk of dying than babies born to mothers who do not work. This result conflicts with those of an Ethiopian study, which showed that the risks of under-five mortality were lower among children born to mothers who did not work than among mothers who were employed in the agricultural sector [21]. Because moms in the agricultural sector have higher incomes than mothers who aren't working, this discrepancy in results may be explained by differences in the socioeconomic and demographic status of the mothers. In the current study, the important risk factors for under-five-year-old children's time to death were shown to grow as their mother's age increased. This result is in line with those of earlier research carried out in Sudan [25] and Suriname [26]. This may be related to the fact that older moms are more likely to experience pregnancy-related hazards, such as poor neonatal outcomes, according to research [27]. This study, however, contrasted with one conducted in Nigeria [28]. Interventions must therefore be coordinated in order to decrease dangerous conditions like pregnancy among older mothers. Another key socio-demographic factor that affects the time to death among under-five children is their place of residence. Compared to urban people, the children of rural residents were more likely to die before the age of five. This result is in line with research done in Ethiopia [29–31]. Finally, this study demonstrated that children who were not immunized had

a higher risk of dying sooner than those who were. This result is in line with research done in Ethiopia [11].

## Limitation of the study

The EDHS is largely based on respondents' self-reports, which raises the possibility of recall bias given the interview's retrospective study design and the high number of missing values in the 2016 EDHS data. Moreover, some significant predictor variables, such as the gestational mother's age, were left out of the study. The investigators should be advised that when their dataset is structure in hierarchical, as in the 2016 EDHS data set, multilevel model analysis is preferable totter compensate for these limitations. Moreover, this investigation was done seven years ago so it is unlikely to reflect the latest status of the mortality rate in the pastoral region of Ethiopia.

## Conclusions

In this study, the authors found a higher rate of under-five deaths than the national figure. A child vaccinated, exclusively breastfeeding, mother's occupation, sex of household head, anemic child, mother's age, and residence were found to be the most influential predictors for time-to-death. Therefore, to lower the high incidence of under-five mortality, the government should focus on the pastoral regional states of Ethiopia.

## Supporting information

**S1 Table. Survival experience comparison.**
(DOCX)

**S1 Appendix.**
(DOCX)

## Acknowledgments

We greatly acknowledge MEASURE DHS for permitting access to the 2016 EDHS data set.
    Moreover, both authors wish to thanks a lot the anonymous reviewers and editor for their several insightful and constructive comments.

## Author Contributions

**Conceptualization:** Bsrat Tesfay Hagos, Gebru Gebremeskel Gebrerufael.

**Data curation:** Bsrat Tesfay Hagos.

**Formal analysis:** Bsrat Tesfay Hagos.

**Methodology:** Bsrat Tesfay Hagos, Gebru Gebremeskel Gebrerufael.

**Software:** Bsrat Tesfay Hagos, Gebru Gebremeskel Gebrerufael.

**Supervision:** Bsrat Tesfay Hagos, Gebru Gebremeskel Gebrerufael.

**Validation:** Gebru Gebremeskel Gebrerufael.

**Writing – original draft:** Bsrat Tesfay Hagos, Gebru Gebremeskel Gebrerufael.

**Writing – review & editing:** Bsrat Tesfay Hagos, Gebru Gebremeskel Gebrerufael.

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
