## [Decision Letter · Decision Letter 0]

27 Jun 2023

PONE-D-23-08757Time to death predictors among under-five children in pastoral regions of Ethiopia: a cross-sectional studyPLOS ONE

Dear Dr. Hagos,

Thank you for submitting your manuscript to PLOS ONE. After careful consideration, we feel that it has merit but does not fully meet PLOS ONE’s publication criteria as it currently stands. Therefore, we invite you to submit a revised version of the manuscript that addresses the points raised during the review process.

Reviewer #1: The authors can find the recommended comments that will support them to imptove the final writing. Technically, even its is soundable, i asked them why they tried to do thiss research with the presence of other similar findings. Likewize i recomend the author to conduct metanalysis instead of this primary study.

Time to death predictors among under-five children in pastoral regions of Ethiopia: a cross-sectional study

Title is well and informative

Abstract

Is well written, but why Italized?

Background

Is good, but needs some modification in logical flow and English writing improvevents

Objective

Its better if it is included in the background section as a last paragraph

In addition, its not recommended to explain about predictors before talking about burden of the issue. Hence rearrangement is mandatory

Method

In lines 90,91 “This study has used an institutionally-based retrospective study and secondary data analysis based on the 2016 Ethiopia Demographic and Health Survey (EDHS).” Indicates as the authors used two different datasets. But the analysis did not indicated it

Hence there is a methodological confussions, needs clarification for the readers

On line 111 & 112 “The response variable for this study was considered to be the length (survival) of time measured in months from date of birth until time-to-death (censor)”, time to death is indicated as censor. But how it could be?

Table 1 (operational definitions) part is not necessary

Results

Even it is good there are some points which are vegue to understand such as “Accordingly, the critical risk factors of time-to-death among children who were born to mothers with advanced age were and 2.18 times higher as compared to their reference (Adjusted Hazard Ratio (AHR) = 1.27; 95% CI: 1.06, 1.52) and (AHR = 2.18; 95% CI:1.59, 2.98), respectively.”

General

What is the implications of your findings?

What new findings you bring?

Why the authors used 2016 EDHS data? Why not used the nearest dataset which is 2019?

In scientific writing, its is not recommended to use pronounce such as we, our,…. So  please replace with appropriate replaces.

 Reviewer #2: Dear Authors, thank you for your commitment and hard work in producing this manuscript with important research topic. I have tried to see your manuscript entitled “Time to death predictors among under-five children in pastoral regions of Ethiopia. A cross-sectional study “and found serious typo errors, significant text overlap with previously published works. Hence, it needs through edition. I will be happy if I get clarification on the following concerns:

1. What is time to death? How did you determine it?

2. Is it cross sectional or follow up study? What do you mean by institutional based retrospective cross-sectional study in abstract session? IF EDHS, so how??

3. Was date of death recorded on the DHS data? If not how can you say time to death?

4. Do you think your data is survival data? If so, why don’t determine median time of death and incidence of death than prevalence? If not, how do you see using survival data analysis…cox regression, PHA…and so on?

5. Don’t you think old data to estimate the current under-five mortality? EDHS 2016???

We look forward to receiving your revised manuscript.

Kind regards,

Mohammed Feyisso Shaka, MPH

Academic Editor

PLOS ONE

Journal Requirements:

2. PLOS requires an ORCID iD for the corresponding author in Editorial Manager on papers submitted after December 6th, 2016. Please ensure that you have an ORCID iD and that it is validated in Editorial Manager. To do this, go to ‘Update my Information’ (in the upper left-hand corner of the main menu), and click on the Fetch/Validate link next to the ORCID field. This will take you to the ORCID site and allow you to create a new iD or authenticate a pre-existing iD in Editorial Manager. Please see the following video for instructions on linking an ORCID iD to your Editorial Manager account: https://www.youtube.com/watch?v=_xcclfuvtxQ.

Reviewers' comments:

Reviewer's Responses to Questions

**Comments to the Author**

1. Is the manuscript technically sound, and do the data support the conclusions?

Reviewer #1: Yes

Reviewer #2: No

2. Has the statistical analysis been performed appropriately and rigorously? 

Reviewer #1: Yes

Reviewer #2: No

3. Have the authors made all data underlying the findings in their manuscript fully available?

Reviewer #1: No

Reviewer #2: Yes

4. Is the manuscript presented in an intelligible fashion and written in standard English?

Reviewer #1: No

Reviewer #2: No

5. Review Comments to the Author

Reviewer #1: The authors can find the recommended comments that will support them to imptove the final writing. Technically, even its is soundable, i asked them why they tried to do thiss research with the presence of other similar findings. Likewize i recomend the author to conduct metanalysis instead of this primary study.

Reviewer #2: Dear Authors, thank you for your commitment and hard work in producing this manuscript with important research topic. I have tried to see your manuscript entitled “Time to death predictors among under-five children in pastoral regions of Ethiopia. A cross-sectional study “and found serious typo errors, significant text overlap with previously published works. Hence, it needs through edition. I will be happy if I get clarification on the following concerns:

1. What is time to death? How did you determine it?

2. Is it cross sectional or follow up study? What do you mean by institutional based retrospective cross-sectional study in abstract session? IF EDHS, so how??

3. Was date of death recorded on the DHS data? If not how can you say time to death?

4. Do you think your data is survival data? If so, why don’t determine median time of death and incidence of death than prevalence? If not, how do you see using survival data analysis…cox regression, PHA…and so on?

5. Don’t you think old data to estimate the current under-five mortality? EDHS 2016???

6. PLOS authors have the option to publish the peer review history of their article (what does this mean?). If published, this will include your full peer review and any attached files.

Reviewer #1: No

Reviewer #2: No

---

## [Author Response · Author response to Decision Letter 0]

16 Jul 2023

Author’s response to reviews

Title: Time to death predictors among under-five children in pastoral regions of Ethiopia: a retrospective follow-up study 

Authors:

 Gebru Gebremeskel Gebrerufael (gebrugebremeskel12@gmail.com) | ORCID: https://orcid.org/0000-0003-2795-8529

Bsrat Tesfay Hagos (bsrattesfaygu@gmail.com) | ORCID: https://orcid.org/0009-0001-0487-7577

Version: 1 Date: 7 July 2023

Author’s response to reviews:

7 July 2023

Response to editors

Dear editors-in-chief,

The authors have enclosed a revised version of manuscript in PLOS ONE research article format, entitled “Time to death predictors among under-five children in pastoral regions of Ethiopia: a retrospective follow-up study”. In this revised version of manuscript, the authors have tried to address all of the editors’ comments, as described in detail in the Response to Editors Comments on the following pages.

As described in my cover letter accompanying this original revised version of manuscript submission, the authors believe that this revised version of manuscript fits very well within the scope of PLOS ONE, as it covers a number of computational approaches and issues that are dominant to the analysis of public health issues.

With warm regards, 

Bsrat Tesfay Hagos

On the behalf of the other authors

Response to Editor and Reviewers comments:

Title: Time to death predictors among under-five children in pastoral regions of Ethiopia: a retrospective follow-up study 

Authors: Gebru Gebremeskel Gebrerufael and Bsrat Tesfay Hagos 

MS ID: PONE-D-23-08757

Journal: PLOS ONE

Article type: Research article

The authors would like to thank you so much for the anonymous editors and reviewers for their precious time and understanding review of this manuscript, and for all of their supportive comments. They raise significant concerns and their inputs are very supportive for improving the main manuscript. The authors agree with almost all their comments and have revised this main manuscript accordingly. Provided underneath is a point-by-point response in purple colored describing by the authors attempts to solve all of the requested revisions in our main manuscript.

Editor’s comments:

Journal Requirements:

Comment 1: Please ensure that your manuscript meets PLOS ONE's style requirements, including those for file naming. The PLOS ONE style templates can be found at https://journals.plos.org/plosone/s/file?id=wjVg/PLOSOne_formatting_sample_main_body.pdf and https://journals.plos.org/plosone/s/file?id=ba62/PLOSOne_formatting_sample_title_authors_affiliations.pdf.

Response 1: Thank you very much editors. The authors really highly appreciate for these constructive comments and the authors have followed the PLOS ONE's style requirements in order to meet its format. 

Comment 2: PLOS requires an ORCID iD for the corresponding author in Editorial Manager on papers submitted after December 6th, 2016. Please ensure that you have an ORCID iD and that it is validated in Editorial Manager. To do this, go to ‘Update my Information’ (in the upper left-hand corner of the main menu), and click on the Fetch/Validate link next to the ORCID field. This will take you to the ORCID site and allow you to create a new iD or authenticate a pre-existing iD in Editorial Manager. Please see the following video for instructions on linking an ORCID iD to your Editorial Manager account: https://www.youtube.com/watch?v=_xcclfuvtxQ.

Response 2: Thank you so much for your insightful comments. The corresponding author creates an ORCID: https://orcid.org/0009-0001-0487-7577. See on Title page 1 and line number 11-12.

Comment 3: Your ethics statement should only appear in the Methods section of your manuscript. If your ethics statement is written in any section besides the Methods, please delete it from any other section.

Response 3: Thank you very much editor. The authors really appreciated for this valuable comment and deleted the unnecessary ethics statement which was written in the “Declaration section” of the main manuscript. 

Reviewer #1

Comment 1: Title is well and informative

Response 1: Thanks so much Reviewer #1. We really highly appreciate for your encouragement.

Abstract

Comment 2: Is well written, but why Italized?

Response 2: Thank you very much for your insightful and fruitful feedback Reviewer #1. The authors really appreciate for this valuable comment and have removed the unnecessary italic style according your suggestion in the main revised version of the manuscript. See Abstract section page 1 and line number 13-40.

Background

Comment 3: Is good, but needs some modification in logical flow and English writing improvevents

Response 3: Thank you so much reviewer for your perceptive and constructive comments. The authors have revised the whole part of the main revised version of the manuscript which is corrected in purple colored. 

Objective

Comment 4: Its better if it is included in the background section as a last paragraph. In addition, it’s not recommended to explain about predictors before talking about burden of the issue. Hence rearrangement is mandatory.

Response 4: Thank you a lot Reviewer, the authors moved the “objective of the study” in to the last paragraph of the introduction/Background section. See page 3-4 and line number 75-85. 

Method

Comment 5: In lines 90, 91 “This study has used an institutionally-based retrospective study and secondary data analysis based on the 2016 Ethiopia Demographic and Health Survey (EDHS).” Indicates as the authors used two different datasets. But the analysis did not indicated it. Hence there is a methodological confussions, needs clarification for the readers

Response 5: Thank you so much for your insightful comments. The authors rewrite the given statement again as suggested. See page 4 and line number 94-95. 

Comment 6: On line 111 & 112 “The response variable for this study was considered to be the length (survival) of time measured in months from date of birth until time-to-death (censor)”, time to death is indicated as censor. But how it could be? 

Response 6: Thank you very much reviewer. The authors revised it in the main manuscript according your suggestion. See page 6 and line number 117-120.

Comment 7: Table 1 (operational definitions) part is not necessary

Response 7: Thank you so much reviewer. The authors revised it in the main manuscript according your recommendation. See page 6 and line number 120-130.

Results

sComment 8: Even it is good there are some points which are vegue to understand such as “Accordingly, the critical risk factors of time-to-death among children who were born to mothers with advanced age were and 2.18 times higher as compared to their reference (Adjusted Hazard Ratio (AHR) = 1.27; 95% CI: 1.06, 1.52) and (AHR = 2.18; 95% CI:1.59, 2.98), respectively.”

Response 8: Thank you Reviewer. The authors highly appreciated your insightful comments. The authors corrected it in the main revised version of the manuscript according your constructive suggestion. See page 13 and line number 216-218.

General 

Comment 9: What are the implications of your findings?

Response 9: Thank you, reviewer, for the valuable comments. The implications for these findings were already given in the “Significance of the study”. See page 4 and line number 86-91.

Comment 10: What new findings you bring?

Response 10: I appreciate your thoughts as well. Therefore, the new findings are given in “strength of the study”. See page 18 and line number 300-306.

Comment 11: Why the authors used 2016 EDHS data? Why not used the nearest dataset which is 2019?

Response 11: Thank you very much Reviewer. The authors highly appreciated your insightful and constructive suggestions. However, this study was conducted before the current 2019 dataset released (since 2021). At that time there was no internet connection in Tigray regional state due to the war crises between regional (Tigray) and federal government (Ethiopia). Therefore, the authors should include this in the limitation of the study. See page 18 and line number 313 - 315.

Comment 12: In scientific writing, its is not recommended to use pronounce such as we, our,…. So please replace with appropriate replaces.

Response 12: Thanks a lot reviewer. The authors highly appreciated this perceptive and fruitful suggestion. Therefore, the authors corrected the whole part on the revised version of the manuscript.

Reviewer #2: 

Dear Authors, thank you for your commitment and hard work in producing this manuscript with important research topic. I have tried to see your manuscript entitled “Time to death predictors among under-five children in pastoral regions of Ethiopia. A retrospective follow-up study “and

Comment 1: I found serious typo errors, significant text overlap with previously published works. Hence, it needs through edition. I will be happy if I get clarification on the following concerns:

Response 1: Thank you very much Reviewer #2. The authors highly appreciated your insightful and constructive suggestions. Therefore, the authors modified the overall parts of the revised version of the manuscript like the typo errors and overlap text of the main manuscript according your suggestions. 

Comment 2: What is time to death? How did you determine it?

Response 2: Thank you very much Reviewer #2. The authors really appreciate for this valuable comment. Therefore, “the child's time to death, which was measured in months from birth to the end of the follow-up period, was the response variable. Events were defined as child deaths between birth and 59 months (1 = died). Children who were still alive at the end of the follow-up period and had not turned five were classified as censored (0 = censored)”. See page 6 and line number 117 - 120.

Comment 3: Is it cross sectional or follow up study? What do you mean by institutional based retrospective cross-sectional study in abstract session? IF EDHS, so how??

Response 3: Thank you so much reviewer for your suggestions. Therefore, the authors corrected as your constructive suggestion. So, it is a retrospective follow-up study design which is the time from date of birth to an event, which can be measured in months and also categorized into Censored (Alive) and Death”.

Comment 4: Was date of death recorded on the DHS data? If not how can you say time to death?

Response 4: Thank you very much Reviewer. Alright! Time from child birth to death and time from child birth to end follow-up time for alive (survival time in months) was recorded in the 2016 EDHS datasets. This, “time to death, which was measured in months from birth to the end of the follow-up period, was the response variable. Events were defined as child deaths between birth and 59 months (1 = died). Children who were still alive at the end of the follow-up period and had not turned five were classified as censored (0 = censored)”. 

Comment 5: Do you think your data is survival data? If so, why don’t determine median time of death and incidence of death than prevalence? If not, how do you see using survival data analysis…cox regression, PHA…and so on?

Response 5: Thank you a lot Reviewer. Yes, it is survival data. Because it has its own retrospective follow-up time from the date of child birth until the event occurred (Dead or Censored). The mean follow-up period was 9 months. The incidence of death rate was 8.4% (95% CI 7.77%, 9.0%). See on “Abstract section” page 1 and line number 26 - 27. Moreover, this is written in the “Result section”.

Comment 6: Don’t you think old data to estimate the current under-five mortality? EDHS 2016???

Response 6: Thank you so much reviewer for your suggestions. However, this comment is already given by another reviewer and the authors have explained briefly. See on the “Limitation of the study” page 18 and line number 313 - 315.

---

## [Decision Letter · Decision Letter 1]

25 Jan 2024

PONE-D-23-08757R1Time to death predictors among under-five children in pastoral regions of Ethiopia: a retrospective follow-upstudyPLOS ONE

Dear Dr. Hagos,

Thank you for submitting your manuscript to PLOS ONE. After careful consideration, we feel that it has merit but does not fully meet PLOS ONE’s publication criteria as it currently stands. Therefore, we invite you to submit a revised version of the manuscript that addresses the points raised during the review process.

We look forward to receiving your revised manuscript.

Kind regards,

Mohammed Feyisso Shaka, MPH

Academic Editor

PLOS ONE

Journal Requirements:

Additional Editor Comments:

Please find more review comments attached below.

Reviewers' comments:

Reviewer's Responses to Questions

Comments to the Author

1. If the authors have adequately addressed your comments raised in a previous round of review and you feel that this manuscript is now acceptable for publication, you may indicate that here to bypass the “Comments to the Author” section, enter your conflict of interest statement in the “Confidential to Editor” section, and submit your "Accept" recommendation.

Reviewer #1: All comments have been addressed

2. Is the manuscript technically sound, and do the data support the conclusions?

Reviewer #1: Yes

3. Has the statistical analysis been performed appropriately and rigorously? 

Reviewer #1: Yes

4. Have the authors made all data underlying the findings in their manuscript fully available?

Reviewer #1: Yes

5. Is the manuscript presented in an intelligible fashion and written in standard English?

Reviewer #1: Yes

6. Review Comments to the Author

Reviewer #1: All the comments found in the attached file. The authors can find and address the final recommendation stated in the attached file.

7. PLOS authors have the option to publish the peer review history of their article (what does this mean?). If published, this will include your full peer review and any attached files.

Do you want your identity to be public for this peer review? For information about this choice, including consent withdrawal, please see our Privacy Policy.

Reviewer #1: Yes: Wolde Melese Ayele

---

## [Author Response · Author response to Decision Letter 1]

17 Feb 2024

Author’s response to reviews

Title: Time to death predictors among under-five children in pastoral regions of Ethiopia: a retrospective follow-up study 

Authors:

 Bsrat Tesfay Hagos (bsrattesfaygu@gmail.com) | ORCID: https://orcid.org/0009-0001-0487-7577

Gebru Gebremeskel Gebrerufael (gebrugebremeskel12@gmail.com) | ORCID: https://orcid.org/0000-0003-2795-8529

Version: 2 Date: 17-Feb-2024

Author’s response to reviews:

Response to editors

Dear editors-in-chief,

The authors have been enclosed a revised version of the manuscript in PLOS ONE research article format, entitled “Time to death predictors among under-five children in pastoral regions of Ethiopia: a retrospective follow-up study." As detailed in the Response to Editors Comments on the following pages, the authors have attempted to address every one of the editors' comments in this revised version of the manuscript.

The authors feel that this revised version of the manuscript fits very well within the scope of PLOS ONE because it covers a number of computational approaches and issues that are important to the analysis of public health issues, as stated in my cover letter that went along with this original revised version of the manuscript submission.

With warm regards, 

Bsrat Tesfay Hagos

On the behalf of the other authors

Response to Editor and Reviewers comments:

Title: Time to death predictors among under-five children in pastoral regions of Ethiopia: a retrospective follow-up study 

Authors: Gebru Gebremeskel Gebrerufael and Bsrat Tesfay Hagos 

MS ID: PONE-D-23-08757R1

Journal: PLOS ONE

Article type: Research article

The authors would like to thank you so much to the anonymous editors and reviewers for their precious time and understanding review of this manuscript and for all of their supportive and constructive comments. They raise significant concerns, and their input is very supportive of improving the main manuscript. The authors agree with almost all their comments and have revised this main manuscript accordingly. The authors provided underneath a point-by-point response in purple colored, which describes attempts to solve all of the requested revisions in our main manuscript.

Editor’s comments:

Comment 1: Abstract

In the background section of the abstract, there is a controversial ideas. The statement indicated that there is a regional variation of the rate of survival among under five years children, which indicated that there are multiple studies conducted. Hence, is it beneficial to reproduce the other findings? Logical answer might be needed

Response 1: Thank you a lot. We authors have amended according your constructive comments. See on the background Abstract section.

Comment 2: Objective

According to the plos guideline, subsection of objective/s is not necessary. It is better to include in the instruction section as a last closing paragraph. Thank you authors since you considered all the comments in your revision

Response 2: Thank you very much. The authors really highly appreciate for these constructive comments and also amended according the PLOS ONE's style requirements in order to meet its format. See page 04 and line#75-79. 

Comment 3: Ethical issue

I recommend attaching the consent approval letter with respect to the title as a link or other supporting information.

Response 3: Thank you so much for your insightful comments. The authors uploaded as a supplementary material about the Ethical issues.

Comment 4: Result

This is my general recommendation: please present the findings as what you found. Don’t use comparative words like more, majority, …. Its better if you use them in the interpretation section(discussion).

Response 4: Thank you so much reviewer for your perceptive and constructive comments. The authors have revised the whole part of the main revised version of the manuscript which is corrected in Page 08 and line#170-171 and 174-175. 

Comment 5: Change all P-values presented as 0.000 to 0.001

Response 5: Thank you very much. We authors have amended all over the manuscript and really highly appreciate for these constructive comments.

Comment 6: Conclusion

On line 320, the authors concluded as “Therefore, to lower the high incidence of under-five mortality, the government should focus on the pastoral regional states of Ethiopia”, but how did you dicide that this high mortality figure is observed in only pastoral regions based on the current study?

Response 6: Thank you very much, reviewer. According to the EDHS 2016 report, the highest rate was recorded in Somalia, Benishangul-Gumuz, and Afar national regional states, where it was estimated to be about 94, 98, and 125 deaths per 1,000 live births, respectively. However, this is still higher than the average national level (Ethiopia) (i.e., reported to be 67 deaths per 1,000 live births).

General recommendation to the authors

Recommendation 1: There are grammatical errors that need dataile revision

Response 1: Thank you Reviewer. The authors highly appreciated your insightful comments. The authors have corrected it in the main revised version of the manuscript according your constructive suggestion.

Recommendation 2: Logical fallacy of some order of presentation of the data

Response 2: Thank you so much reviewer. The authors revised it in the main manuscript according your recommendation.

Recommendation 3: Better if the study was included all Ethiopian regions and use of multilevel analysis. This might have significant conclusion and greater practical implication

Response 3: Thank you very much Reviewer. The authors have highly appreciated for your impressive and constructive suggestions. See page 18 and line#316-318.

Recommendation 4: Overall the authors atached all previous recommendation, so accepted the manuscript with a minor recommendations.

Response 4: Thanks a lot reviewer. The authors have extremely respected this understanding and fruitful suggestion. Therefore, the authors corrected the whole part on the revised version of the manuscript. And also thanks so much for your encouragement.

---

## [Editor Report · Decision Letter 2]

2 Apr 2024

PONE-D-23-08757R2Time to death predictors among under-five children in pastoral regions of Ethiopia: a retrospective follow-upstudyPLOS ONE

Dear Dr. Hagos,

Thank you for submitting your manuscript to PLOS ONE. After careful consideration, we feel that it has merit but does not fully meet PLOS ONE’s publication criteria as it currently stands. Therefore, we invite you to submit a revised version of the manuscript that addresses the points raised during the review process.

Your manuscript needs further revision. Please find the comments in the track change on the file attached and address them thoroughly.  ==============================

We look forward to receiving your revised manuscript.

Kind regards,

Mohammed Feyisso Shaka, MPH

Academic Editor

PLOS ONE
---

## [Author Response · Author response to Decision Letter 2]

16 Apr 2024

Author’s response to reviews

Title: Predictors of time to death among under-five children in pastoral regions of Ethiopia: a retrospective follow-up study 

Authors:

 Bsrat Tesfay Hagos (bsrattesfaygu@gmail.com) | ORCID: https://orcid.org/0009-0001-0487-7577

Gebru Gebremeskel Gebrerufael (gebrugebremeskel12@gmail.com) | ORCID: https://orcid.org/0000-0003-2795-8529

Brhane Gebrehiwot Welegebrial (brhanegb008@gmail.com) ǀ ORCID: https://orcid.org/0000-0003-3600-7477

Version: 3 Date: 12-April-2024

Author’s response to reviews:

Response to editors

Dear editors-in-chief,

The authors have been enclosed a revised version of the manuscript in PLOS ONE research article format, entitled “Predictors of time to death among under-five children in pastoral regions of Ethiopia: a retrospective follow-up study" As detailed in the Response to Editors Comments on the following pages, the authors have attempted to address every one of the editors' comments in this revised version of the manuscript.

The authors feel that this revised version of the manuscript fits very well within the scope of PLOS ONE because it covers a number of computational approaches and issues that are important to the analysis of public health issues, as stated in my cover letter that went along with this original revised version of the manuscript submission.

With warm regards, 

Bsrat Tesfay Hagos

On the behalf of the other authors

Response to Editor and Reviewers comments:

Title: Predictors of time to death among under-five children in pastoral regions of Ethiopia: a retrospective follow-up study 

Authors: Gebru Gebremeskel Gebrerufael and Bsrat Tesfay Hagos 

MS ID: PONE-D-23-08757R1

Journal: PLOS ONE

Article type: Research article

The authors would like to thank you so much to the anonymous editors and reviewers for their precious time and understanding review of this manuscript and for all of their supportive and constructive comments. They raise significant concerns, and their input is very supportive of improving the main manuscript. The authors agree with almost all their comments and have revised this main manuscript accordingly. The authors provided underneath a point-by-point response in purple colored, which describes attempts to solve all of the requested revisions in our main manuscript.

---

## [Editor Report · Decision Letter 3]

30 Apr 2024

PONE-D-23-08757R3Predictors of time to death among under-five children in pastoral regions of Ethiopia: a retrospective follow-up studyPLOS ONE

Dear Dr. Hagos,

Thank you for submitting your manuscript to PLOS ONE. After careful consideration, we feel that it has merit but does not fully meet PLOS ONE’s publication criteria as it currently stands. Therefore, we invite you to submit a revised version of the manuscript that addresses the issue raised during the last review process.

Please remove the extra author included after the completion of the substantial review of the manuscript. ==============================

We look forward to receiving your revised manuscript.

Kind regards,

Mohammed Feyisso Shaka, MPH

Academic Editor

PLOS ONE
---

## [Author Response · Author response to Decision Letter 3]

6 May 2024

Author’s response to reviews

Title: Predictors of time to death among under-five children in pastoral regions of Ethiopia: a retrospective follow-up study 

Authors:

 Bsrat Tesfay Hagos (bsrattesfaygu@gmail.com) | ORCID: https://orcid.org/0009-0001-0487-7577

Gebru Gebremeskel Gebrerufael (gebrugebremeskel12@gmail.com) | ORCID: https://orcid.org/0000-0003-2795-8529

Version: 4 Date: 02-May-2024

Author’s response to reviews:

Response to editors

Dear editors-in-chief,

The authors have been enclosed a revised version of the manuscript in PLOS ONE research article format, entitled “Predictors of time to death among under-five children in pastoral regions of Ethiopia: a retrospective follow-up study" As detailed in the Response to Editors Comments on the following pages, the authors have attempted to address every one of the editors' comments in this revised version of the manuscript.

The authors feel that this revised version of the manuscript fits very well within the scope of PLOS ONE because it covers a number of computational approaches and issues that are important to the analysis of public health issues, as stated in my cover letter that went along with this original revised version of the manuscript submission.

With warm regards, 

Bsrat Tesfay Hagos

On the behalf of the other authors

Response to Editor and Reviewers comments:

Title: Predictors of time to death among under-five children in pastoral regions of Ethiopia: a retrospective follow-up study 

Authors: Gebru Gebremeskel Gebrerufael and Bsrat Tesfay Hagos 

MS ID: PONE-D-23-08757R3

Journal: PLOS ONE

Article type: Research article

The authors would like to thank you so much to the anonymous editors and reviewers for their precious time and understanding review of this manuscript and for all of their supportive and constructive comments. They raise significant concerns, and their input is very supportive of improving the main manuscript. The authors agree with almost all their comments and have revised this main manuscript accordingly. The authors provided underneath a point-by-point response in purple colored, which describes attempts to solve all of the requested revisions in our main manuscript.

Editor’s comments:

Comment 1: Please remove the extra author included after the completion of the substantial review of the manuscript. 

Response 1: Thank you very much Reviewer. The authors have removed according your best suggestion. 

Comment 2: Please review your reference list to ensure that it is complete and correct. If you have cited papers that have been retracted, please include the rationale for doing so in the manuscript text, or remove these references and replace them with relevant current references. Any changes to the reference list should be mentioned in the rebuttal letter that accompanies your revised manuscript. If you need to cite a retracted article, indicate the article’s retracted status in the References list and also include a citation and full reference for the retraction notice.

Author Response 2: Thank you so much for your constructive and impressive suggestions. Therefore, the authors have corrected according your suggestions overall the revised manuscript. See on the references section on page #18-22 and #line 323-419.

---

## [Editor Report · Decision Letter 4]

16 May 2024

Predictors of time to death among under-five children in pastoral regions of Ethiopia: a retrospective follow-up study

PONE-D-23-08757R4

Dear Dr. Hagos,

We’re pleased to inform you that your manuscript has been judged scientifically suitable for publication and will be formally accepted for publication once it meets all outstanding technical requirements.

Kind regards,

Mohammed Feyisso Shaka, MPH

Academic Editor

PLOS ONE
---

## [Editor Report · Acceptance letter]

25 Jun 2024

PONE-D-23-08757R4 

PLOS ONE

Dear Dr. Hagos, 

I'm pleased to inform you that your manuscript has been deemed suitable for publication in PLOS ONE. Congratulations! Your manuscript is now being handed over to our production team.

Kind regards, 

on behalf of

Mr. Mohammed Feyisso Shaka 

Academic Editor

PLOS ONE